# Photocatalytic Isomerization of (*E*)-Anethole to (*Z*)-Anethole

**DOI:** 10.3390/molecules27165342

**Published:** 2022-08-22

**Authors:** Marvin Korff, Tiffany O. Paulisch, Frank Glorius, Nikos L. Doltsinis, Bernhard Wünsch

**Affiliations:** 1Westfälische Wilhelms-Universität Münster, Chemical Biology of Ion Channels (Chembion), GRK 2515, Corrensstraße 48, D-48149 Münster, Germany; 2Westfälische Wilhelms-Universität Münster, Institut für Pharmazeutische und Medizinische Chemie, Corrensstraße 48, D-48149 Münster, Germany; 3Westfälische Wilhelms-Universität Münster, Organisch-Chemisches Institut, Corrensstraße 40, D-48149 Münster, Germany; 4Westfälische Wilhelms-Universität Münster, Institut für Festkörpertheorie und Center for Multiscale Modelling and Computation, Wilhelm-Klemm-Straße 10, D-48149 Münster, Germany

**Keywords:** catalysis, green chemistry, natural product, photocatalysis, photosensitizer, (*E*/*Z*)-isomerization, diastereoselective dihydroxylation, DFT calculations, calculation of triplet energy

## Abstract

Natural product (*E*)-anethole was isomerized to (*Z*)-anethole in a photocatalytic reaction. For this purpose, a self-designed cheap photoreactor was constructed. Among 11 photosensitizers (organo and metal complex compounds), Ir(p-*t*Bu-ppy)_3_ led to the highest conversion. Triplet energies of (*E*)- and (*Z*)-anethole were predicted theoretically by DFT calculations to support the selection of appropriate photosensitizers. A catalyst loading of 0.1 mol% gave up to 90% conversion in gram scale. Further additives were not required and mild irradiation with light of 400 nm overnight was sufficient. As a proof of concept, (*E*)- and (*Z*)-anethole were dihydroxylated diastereoselectively to obtain diastereomerically pure *like*- and *unlike*-configured diols, respectively.

## 1. Introduction

(*E*)-Anethole is a natural product and is the main component of anise oil (80–99%). It is commonly used as a flavor in foods and has antimicrobial properties [1]. Its isomer (*Z*)-anethole is a natural product as well. However, with a lower natural occurrence, and in contrast to (*E*)-anethole, it was not commercially available during the course of this study.

(*Z*)-Olefins in general are thermodynamically less stable as well as synthetically less accessible than their (*E*)-configured counterparts [2]. Synthetic strategies towards (*Z*)-olefins include the *Wittig* reaction, partial hydrogenation of alkynes and *Suzuki–Miyaura* coupling. In the case of (*Z*)-anethole, the *Wittig* reaction showed selectivity issues since considerable amounts of (*E*)-anethole were formed [3]. The amount of (*Z*)-anethole could only be increased using the carcinogenic solvent hexamethylphosphoramide [4]. Hydrogenation of the corresponding alkyne with a poisoned Ni-catalyst also led to mixtures of (*E*)- and (*Z*)-anethole [5]. Additionally, the starting material is not commercially available and has to be synthesized before use. In this study, we tested the *Suzuki–Miyaura* coupling of (4-methoxyphenyl)boronic acid with (*Z*)-1-bromoprop-1-ene. Although trying different Pd catalysts, this reaction led either to low yields or to low diastereoselectivity (see Appendix A).

The first light-mediated (*E*)-/(*Z*)-isomerization of olefins using photosensitizers was reported in the 1960s, using organic molecules such as acetophenone or benzophenone as catalysts and stilbene as substrate [6]. Later on, the scope of substrates (e.g., diphenylpropenes, piperylene and fumarate/maleate) as well as the used photosensitizers was extended [7]. In the case of secondary and tertiary β-alkylstyrenes, the method descriptions of the experiments are ambiguous and explicit reaction conditions are not available [8,9]. With the recently developed high-performing metal complex photocatalysts and their application in a variety of organic reactions [10,11,12,13], the photocatalyzed (*E*)-/(*Z*)-isomerization was revisited. As examples, allylamines and β-alkylstyrenes were isomerized with Ir(ppy)_3_ with good to excellent conversion rates [2].

The postulated mechanism of action involves excitation of the photosensitizer by light from the singlet ground state (S_0_) into the excited singlet state S_1_. Following intersystem crossing (ISC) leads to the triplet state T_1_ (Figure 1a). Upon the collision of the excited photosensitizer with the substrate, a simultaneous intermolecular exchange of ground-state and excited-state electrons via “Dexter Energy Transfer” occurs (see Figure 1a,b and Strieth-Kalthoff et al. [11]). This transfer leads to a relaxed photosensitizer and an excited substrate. For this excitation, spectral overlap (*J*, Figure 2b) of the photosensitizer and the substrate is required and, therefore, the targeted double bond generally has to be in conjugation with a π-system [14].

The substrate, now itself in the T_1_ state, reaches the global energy minimum via rotation around the former C=C double bond (Figure 2a,b). Subsequent intersystem crossing (ISC) transforms the T_1_ into the S_0_ state, which represents the global energy maximum on the way to the (*E*)- or (*Z*)-isomer. The S_0_ state is able to relax to the ground state forming either the (*E*)- or (*Z*)-diastereomer. The transformation from the (*E*)- to the (*Z*)-isomer is reversible if the excited photosensitizer is also able to transfer its energy to the reaction product via “Dexter Energy Transfer”. The ratio of product: substrate in the equilibrium (photostationary state) is dependent on the ratio of the spectral overlap of the photosensitizer and substrate to the spectral overlap of the photosensitizer and product (Figure 2b). To maximize the reaction conversion, the spectral overlap with the product has to be minimized. For a given substrate, this can be achieved by the selection of a photosensitizer with the desired spectral properties. As the triplet energy state of the (*Z*)-olefin is generally higher (orange curve) than that of the (*E*)-olefin (green curve), a photosensitizer with a lower energy should be selected (blue curve) for an endergonic energy transfer. In this case, the “Dexter Energy Transfer” must occur from higher vibrational/rotational states, T_1_, of the photosensitizer to excite the substrate (Figure 1a).

An easy access to (*Z*)-olefins is valuable in organic and medicinal chemistry, as the double bond can be further functionalized, leading, e.g., to a 1,2-disubstituted saturated system. As such a type of transformations can be conducted in a stereoselective manner, (*E*)- and (*Z*)-olefins give access to different stereoisomers. Various diastereoselective reactions (e.g., *Prileschajew* epoxidation [15], *Upjohn* dihydroxylation [16], or hydroboration–oxidation), as well as enantioselective transformations (*Sharpless* dihydroxylation [17], *Sharpless* aminohydroxylation [18], and *Shi* [19], or *Jacobsen* epoxidation [20]) for olefins are well established.

Figure 3 shows a selection of approved drugs with a 1,2-disubstituted phenylalkane motif, which are structurally related to the corresponding olefin: (−)-ephedrine, an unselective indirect sympathomimeticum (inhibition of noradrenaline reuptake), and the antibiotic chloramphenicol and methylphenidate, central nervous system stimulants.

As (*Z*)-anethole should serve as a precursor in drug discovery, it was selected as the target molecule in this study.

## 2. Results

As discussed above, the selection of a photosensitizer with an appropriate triplet energy state (*E*_T_ (PS)) is essential for the isomerization of (*E*)- to (*Z*)-anethole. Therefore, the photosensitizers used in this study are classified according to their triplet energy (*E*_T_ (PS)). High spectral overlap with the substrate (*E*)-anethole ((*E*)-**1**) and low overlap with the reaction product (*Z*)-anethole ((*Z*)-**1**) is desired (Figure 2b). Consequently, knowledge about the respective excitation energies *E*_T_ ((*E*)-**1**) and *E*_T_ ((*Z*)-**1**) is important. Unfortunately, these values are not available in the literature to the best of our knowledge.

Since the recording of singlet–triplet absorption spectra of most organic compounds is challenging, the triplet energy states of (*E*)- and (*Z*)-anethole were calculated theoretically using established methods [11]. For better mechanistic understanding and decision making, the triplet energy states of (*E*)- and (*Z*)-anethole were determined by density functional theory (DFT) using the B3LYP functional and the 6-31G* basis set (for computational details, see Appendix A). For the 0–0 transition, the calculated triplet energy of (*E*)-anethole is *E*_T_ ((*E*)-**1**) = 58.1 kcal/mol and for (*Z*)-anethole *E*_T_ ((*Z*)-**1**) = 60.8 kcal/mol (see Figure 2).

At first, photosensitizers were selected according to their triplet energy states and screened for their potential to photoisomerize (*E*)- to (*Z*)-anethole. The experiments were conducted in analogy to reported conditions [2]. (*E*)-Anethole, *N*,*N*-diisopropyl-*N*-ethylamine (DIPEA), and the respective photosensitizer were dissolved in CH_3_CN under strict air exclusion since O_2_ acts as a triplet quencher. The mixture was irradiated at 400 nm in a photoreactor at 28 °C overnight. The transformation of (*E*)- to (*Z*)-anethole was determined by the integration of characteristic signals in the respective ^1^H NMR spectra of the crude products.

In total, eleven photosensitizers were employed. Four sensitizers represent organocatalysts and seven metal-based catalysts. The results of the photoisomerization using these catalysts are summarized in Table 1. The entries are ordered according to increasing excitation energy (*E*_T_ (PS)) of the photosensitizers.

Triplet energies of 46.5 kcal/mol and less (entry 2 and 3) of the photosensitizer are too low to achieve excitation of (*E*)-anethole to the triplet state, which results in no conversion into (*Z*)-anethole. [Ir(d*t*bbpy)(ppy)_2_](PF_6_) with *E*_T_ = 49.2 kcal/mol (entry 4) however, already shows a conversion of 85%. This rather small difference in energy (Δ*E*_T_ = 2.7 kcal/mol) indicates a rather hard cut-off of the spectral overlap of the photosensitizer and substrate. The organo-photocatalyst riboflavin, with a comparable triplet energy of 49.9 kcal/mol, however, did not induce any conversion (entry 5). It has been successfully employed in the isomerization of functionalized, electron-deficient double bonds, but a different mechanism of action (dichotomic excitation to the S_1_ and T_1_ state of the substrate) has been postulated for this photosensitizer [21]. The highest conversion of 90% was obtained with Ir(*p*-*t*Bu-ppy)_3_ as the photosensitizer with a triplet energy of 54.5 kcal/mol (entry 7). Further increasing the triplet energy of the photocatalyst leads to a kind of plateau (entry 8, 9), and finally to a decrease in conversion (entry 11). With thioxanthone (entry 12), (*E*)- and (*Z*)-anethole were formed in the ratio 50:50, which indicates, that the excitation energy (63.3 kcal/mol) is high enough to excite both (*E*)- and (*Z*)-anethole in the same amount. The control experiment without the photosensitizer described in entry 1 clearly indicates that a photosensitizer is indeed needed for the isomerization to occur.

The results obtained with the organo-photocatalysts do not directly correlate with their excitation energy. Compared with the metal-based photocatalysts, the amount of formed (*Z*)-anethole is considerably lower with the organo-photocatalysts. In the case of benzil, a major formation of side products was observed (entry 6, see Appendix A for further experiments), even though its triplet energy is almost identical to the triplet energy of the Ir-based catalyst Ir(*p*-*t*Bu-ppy)_3_ (Δ*E*_T_ = 0.3 kcal/mol), which led to the highest conversion of all tested photocatalysts. Although benzil (entry 6) and chrysene (entry 10) have already been applied for the photoisomerization of substituted styrene derivatives, the reported procedure remains difficult to reproduce (there irradiation at *λ* = 366 nm) [8,9].

It is worth emphasizing that with the exception of the reaction catalyzed by benzil (entry 6), the photoisomerization reactions described in Table 1 proceeded in a very clean fashion. The ^1^H NMR spectra of non-purified products showed only signals for (*E*)- and (*Z*)-anethole, signals for side products could not be detected. The solvent and additive were readily removed under reduced pressure and, generally, the photosensitizers were precipitated and separated, exploiting the solubility differences of the catalyst and anethole in *n*-pentane. This work-up procedure led to quantitative yields of mixtures of (*E*)- and (*Z*)-anethole, which were pure enough for further chemical reactions.

Within this series of photocatalysts, the Ir-based catalyst Ir(*p*-*t*Bu-ppy)_3_ provided the highest amount of desired (*Z*)-anethole (90%, Table 1, entry 7). Ir(*p*-*t*Bu-ppy)_3_ has a triplet energy of 54.5 kcal/mol, which is 3.6 kcal/mol lower than the triplet energy of (*E*)-anethole (58.1 kcal/mol). This result is in good agreement with the conceptual approach and underlines the potential and relevance of the theoretical analysis of a system. Based on these promising results, Ir(*p*-*t*Bu-ppy)_3_ was selected for further optimization experiments. In particular, the amount of catalyst required for the transformation and the optimal solvent should be explored. The results of the optimization experiments are summarized in Table 2.

**Table 1 molecules-27-05342-t001:** Screening of photosensitizers regarding their potential to convert (*E*)- into (*Z*)-anethole. Organo-photosensitizers were used in an amount of 20 mol% and metal-based photosensitizers in an amount of 2 mol%, relative to (*E*)-anethole. The transformations were determined by recording 1H NMR spectra of the crude products. 
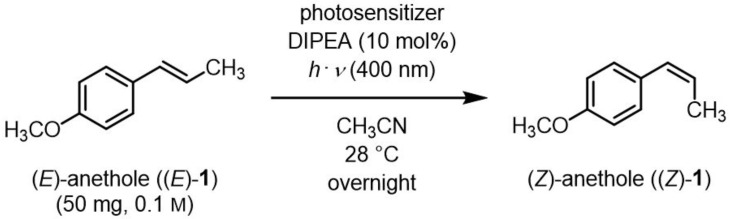
.

Entry	Photosensitizer	*E*_T_ (PS) (kcal/mol)	Ratio (*E*):(Z)
*1*	-	-	100:0
*2*	[Ru(dmbpy)_3_](PF_6_)_2_	45.3 ^c^	100:0
*3*	[Ru(bpy)_3_](BF_4_)_2_	46.5 ^c^	100:0
*4*	[Ir(d*t*bbpy)(ppy)_2_](PF_6_)	49.2 ^c^	15:85
*5 ^a^*	riboflavin	49.9 ^d^	100:0
*6 ^a,b^*	benzil	54.2 ^d^	87:13
*7*	Ir(*p*-*t*Bu-ppy)_3_	54.5 ^e^	10:90
*8*	Ir(ppy)_3_	55.2 ^e^	12:88
*9*	Ir(*p*-CF_3_-ppy)_3_	56.4 ^e^	11:89
*10 ^a^*	chrysene	57.1 ^d^	93:7
*11*	[Ir(dFCF_3_ppy)_2_(d*t*bpy)]PF_6_	60.1 ^e^	26:74
*12*	thioxanthone	63.3 ^d^	50:50

^a^ Without DIPEA. ^b^ Considerable amounts of side products were observed. ^c^ Teegardin et al. [10]. ^d^ Montalti et al. [22]. ^e^ Singh et al. [23]. The given triplet energy values of the catalysts are not absolute values, as they were measured under different conditions. Chemical structures of utilized photosensitizers are given in the Appendix A.

Compared to the control conditions (Table 2, entry 1), doubling the concentration of (*E*)-anethole to 0.2 m and reducing the catalyst amount from 2 mol% to 0.5 mol% or even 0.1 mol% did not influence the conversion (entries 2 and 3). Although DIPEA had been described to positively influence the photoisomerization [2], it did not change the conversion of (*E*)-anethole (entry 4). Changing the solvent from CH_3_CN (entry 4) to CH_2_Cl_2_ (entry 5), THF (entry 6), or CH_3_OH (entry 7) provided the same ratio of (*E*)-anethole: (*Z*)-anethole of 10:90. For reasons of sustainability, methanol was used in all further experiments as it is the environmentally preferred solvent among the four tested [24].

Additionally, efforts to recycle the catalyst were undertaken. At catalyst loadings of 0.1 mol%, it was not possible anymore to remove the photosensitizer by precipitation with *n*-pentane from the crude product. Instead, the catalyst used in entry 7 was separated by column chromatography and isolated in 63% yield. This recovered catalyst was reused in a second photoisomerization reaction leading to the same ratio of (*E*)-anethole: (*Z*)-anethole of 10:90 (entry 8).

Finally, experiments were conducted to identify the time required to reach the photostationary state, which is dependent on the scale of the reaction in relation to the power of the light source, as well as the catalyst loading and its turnover rate. In regular intervals, samples were taken from the reaction mixture and the ratio of (*E*)-anethole: (*Z*)-anethole was determined by ^1^H NMR spectroscopy (see Appendix A). After a period of 8 h, the photostationary state was not reached as only 75% of (*E*)-anethole had been converted into (*Z*)-anethole (entry 9). Obviously, the period “overnight” is the optimal reaction time to bring the transformation to its equilibrium. However, even the overnight period of 22 h led only to 87% conversion, which was explained in this special case by O_2_, which could not be excluded completely when taking the samples in regular intervals. In conclusion, the conditions of the photoisomerization reaction were optimized with regard to catalyst loading (0.1 mol%), the removal of DIPEA additive and employing the environmentally more friendly solvent methanol, but the ratio of (*Z*)-anethole: (*E*)-anethole could not be raised above 90:10.

Since separation of (*E*)-anethole and (*Z*)-anethole via normal or reversed phase column chromatography was not possible, pure (*Z*)-anethole could not be isolated without (*E*)-anethole impurities. However, the ratio of 90:10 did not change upon storage at −20 °C in nitrogen atmosphere for several months, as no isomerization of (*Z*)-anethole to the thermodynamically favored (*E*)-anethole was observed. For quantitative removal of the photosensitizer, distillation is an option, and, depending on the follow-up transformations, the small amounts of (*E*)-anethole and/or derivatives thereof can be easily removed at a later stage of the synthesis.

To demonstrate the value of this photoisomerization reaction for diastereoselective (and potentially even enantioselective) synthetic approaches, (*E*)- and (*Z*)-anethole were oxidized with OsO_4_ in an *Upjohn* dihydroxylation [25]. Due to the *syn*-approach of OsO_4_ to double bonds, (*E*)-anethole provided diastereoselectively *like*-configured diol *like*-**2**, whereas (*Z*)-anethole gave *unlike*-configured diol *unlike*-**2** (Figure 1).

In the case of *unlike*-**2**, impurities of diol *like*-**2** originating from dihydroxylation of the remaining (*E*)-anethole were readily removed by recrystallization from water to obtain pure *unlike*-configured diol *unlike*-**2** in a 62% yield. The remaining trace amounts of the photocatalyst Ir(*p*-*t*Bu-ppy)_3_ in the starting material did not interfere with the dihydroxylation reaction of (*Z*)-anethole ((*Z*)-**1**).

## 3. Conclusions

In this study, a method for the photocatalytic isomerization of (*E*)- to (*Z*)-anethole has been developed. Based on theoretical calculations of triplet energy states of (*E*)-anethole and (*Z*)-anethole, a set of appropriate photocatalysts was selected. Experiments led to the identification of the Ir-based photosensitizer Ir(*p*-*t*Bu-ppy)_3_ giving a high conversion of (*E*)-anethole into (*Z*)-anethole. In general, metal complex catalysts were superior to organo-photocatalysts for this transformation. The amount of the photocatalyst Ir(*p*-*t*Bu-ppy)_3_ could be reduced to 0.1 mol% in methanol without any additive, resulting in a ratio of (*E*)-anethole: (*Z*)-anethole of 10:90. This photoisomerization of (*E*)-anethole to (*Z*)-anethole is a simple transformation of a commercially available, cheap, natural product using a commercially available photosensitizer. The self-made photoreactor can be easily produced using cheap materials. The mild irradiation with 400 nm on the edge between visible and UV light led to a clean transformation without the formation of side products with a conversion of 90%. The catalyst could be separated easily by precipitation with *n*-pentane or column chromatography. The method is very robust against various reaction conditions, is highly atom economic and scalable in gram scale for synthetic applications. (*E*)-anethole and (*Z*)-anethole were oxidized with OsO_4_ yielding, selectively, diols *like*-**2** and *unlike*-**2**, respectively.

It is speculated that this method could be used to isomerize further (*E*)-configured styrene derivatives to obtain less accessible (*Z*)-configured olefins and derivatives thereof.

## 4. Experimental Part

### 4.1. General Procedure for Photoisomerization Experiments

A reaction tube was filled with the respective solvent and degassed in an ultrasonic bath for 10 min and further 10 min in an N_2_ stream while stirring vigorously with a magnetic stir bar. Under N_2_ atmosphere, the photosensitizer, the substrate (*E*)-anethole ((*E*)-**1**) and the additive *N*,*N*-diisopropyl-*N*-ethylamine (DIPEA) were added in the respective order to the reaction vessel. The test tube was sealed with a rubber septum and placed with a clamp into the photoreactor. While stirring, the reaction mixture was irradiated with a wavelength of 400 nm overnight. During irradiation, the temperature was adjusted with the implemented fan to constant 28 °C. The reaction was terminated by switching off the lights and the solvent and additive were removed *in vacuo*. In the case of higher catalyst loadings, *n*-pentane was added to the residue to precipitate the photosensitizer. The suspension was filtrated over a pad of cotton and the solvent was removed *in vacuo*. A sample of the crude product was dissolved in deuterated DMSO and submitted for recording a ^1^H NMR spectrum. Explicit values for reactants’ concentrations and equivalents for a given experiment, as well as the determined conversion, can be found in Table 1 and Table 2.

### 4.2. (Z)-1-Methoxy-4-(prop-1-en-1-yl)benzene ((Z)-anethole, (Z)-1)

The synthesis was performed according to the general procedure for photoisomerization experiments employing (*E*)-anethole (1.00 g, 6.75 mmol), the photocatalyst Ir(*p*-*t*Bu-ppy)_3_ (5.5 mg, 6.8 µmol, 0.1 mol%) and CH_3_OH (34 mL). The crude product was purified by automated flash column chromatography (cyclohexane/EtOAc = 99:1 → 80:20). The title compound was obtained as a colorless liquid (897 mg, 6.05 mmol, 90%), as a mixture of (*Z*)- and (*E*)-anethole in the ratio of (*Z*)-/(*E*)- = 9:1. TLC: *R*_f_ = 0.31 (cyclohexane). ^1^H NMR (600 MHz, DMSO-d_6_): *δ* [ppm] = 1.81* (dd, *J* = 6.6/1.7 Hz, 0.3H, CHCH*_3_*), 1.83 (dd, *J* = 7.2/1.8 Hz, 2.7H, CHCH_3_), 3.73* (s, 0.3H, OCH_3_), 3.75 (s, 2.7H, OCH_3_), 5.66 (dq, *J* = 11.6/7.2 Hz, 0.9H, CH=CHCH_3_), 6.11* (dq, *J* = 15.8/6.6 Hz, 0.1H, CH=CHCH_3_), 6.31–6.38* (m, 1H, CH=CHCH_3_), 6.83–6.88* (m, 0.2H, 2-H/6-H_aryl_), 6.89–6.95 (m, 1.8H, 2-H/6-H_aryl_), 7.22–7.32 (m, 1.8H, 3-H/5-H_aryl_), 7.32–7.27 (m, 0.2H, 3-H/5-H_aryl_). ^13^C NMR (151 MHz, DMSO-d_6_): *δ* [ppm] = 14.4 (0.9C, CHCH_3_), 18.2* (0.1C, CHCH_3_), 55.0 (1C, OCH_3_), 113.7 (1.8C, C-2/C-6_aryl_), 113.9* (0.2C, C-2/C-6_aryl_), 122.9* (0.1C, CH=CHCH_3_), 124.5 (0.9C, CH=CHCH_3_), 126.8* (0.2C, C-3/C-5_aryl_), 129.1 (0.9C, CH=CHCH_3_), 129.6 (0.9C, C-4_aryl_), 129.8 (1.8C, C-3/C-5_aryl_), 130.1* (0.1C, C-4_aryl_), 130.3* (0.1C, CH=CHCH_3_), 157.8 (0.9C, C-1_aryl_), 158.3* (0.1C, C-1_aryl_). Signals marked with * belong to the minor diastereomer (*E*)-anethole. IR (neat): ν (cm^−1^) = 3013 (C–H_aryl_), 2932, 2835 (C–H_alkyl_), 1604, 1508 (C=C_aryl_), 1246 (C–O), 837 (C–H_aryl_). HRMS: *m*/*z* = 149.0973, calcd. 149.0961 for C_10_H_13_O^+^ ([M+H]^+^).

### 4.3. (1. RS,2RS)-1-(4-Methoxyphenyl)propane-1,2-diol (like-2)

(*E*)-Anethole ((*E*)-**1**, 100 mg, 675 µmol) was dissolved in a mixture of acetone (5 mL) and *t*BuOH (2 mL). A mixture of OsO_4_ in 0.05 m H_2_SO_4 aq._ (9.8 mM, 1.7 mL, 17 µmol, 2.5 mol%) and *N*-methylmorpholine-*N*-oxide (NMO, 87.0 mg, 742 µmol, 1.10 eq.) were added to the solution. The mixture was stirred overnight at room temperature. A saturated aqueous solution of Na_2_SO_3_ (10 mL) was added and the mixture was extracted with EtOAc (3 × 10 mL). The organic layers were combined, dried (Na_2_SO_4_) and the solvent was removed *in vacuo*. The product was obtained as colorless solid without further purification, mp 64 °C (Lit. 62–63 °C [26]), yield 118 mg (0.65 mmol, 96%). TLC: *R*_f_ = 0.49 (CH_2_Cl_2_/CH_3_OH = 92:8). ^1^H NMR (600 MHz, DMSO-d_6_): *δ* (ppm) = 0.80 (d, *J* = 6.3 Hz, 3H, CHOHC*H*_3_), 3.60 (quint/d, *J* = 6.3/4.2 Hz, 1H, C*H*OHCH_3_), 3.73 (s, 3H, OCH_3_), 4.21 (dd, *J* = 6.5/3.9 Hz, 1H, C*H*OHCHOHCH_3_), 4.53 (d, *J* = 4.2 Hz, 1H, CHO*H*CH_3_), 5.03 (d, *J* = 3.9 Hz, 1H, CHO*H*CHOHCH_3_), 6.83–6.88 (m, 2H, 2-H/6-H_aryl_), 7.19–7.24 (m, 2H, 3-H/5-H_aryl_). ^13^C NMR (151 MHz, DMSO-d_6_): *δ* (ppm) = 18.7 (1C, CHOH*C*H_3_), 55.0 (1C, OCH_3_), 70.7 (1C, *C*HOHCH_3_), 77.2 (1C, *C*HOHCHOHCH_3_), 113.0 (2C, C-2/C-6_aryl_), 128.1 (2C, C-3/C-5_aryl_), 134.9 (1C, C-4_aryl_), 158.2 (1C, C-1_aryl_). IR (neat): *ν* (cm^−1^) = 3240 (O–H), 2997, 2898 (C–H_alkyl_), 1609, 1512 C=C_aryl_), 1242, 1030 (C–O), 829 (C–H_aryl_). HRMS: *m*/*z* = 165.0929, calcd. 165.0910 for C_10_H_13_O_2_^+^ [M+H-H_2_O]^+^.

### 4.4. (1. RS,2SR)-1-(4-Methoxyphenyl)propane-1,2-diol (unlike-2)

(*Z*)-Anethole ((*Z*)-**1**, 1.00 g, 6.75 mmol, containing 10% (*E*)-anethole) was dissolved in a mixture of acetone (50 mL) and *t*BuOH (20 mL). A mixture of OsO_4_ in 0.05 m H_2_SO_4 aq._ (9.8 mM, 17 mL, 0.17 mmol, 2.5 mol%) and *N*-methylmorpholine-*N*-oxide (NMO, 869 mg, 7.42 mmol, 1.10 eq.) were added to the solution. The mixture was stirred overnight at room temperature. A saturated aqueous solution of Na_2_SO_3_ (50 mL) and H_2_O (100 mL) was added and the mixture was extracted with EtOAc (3 × 100 mL). The organic layers were combined and dried (Na_2_SO_4_), and the solvent was removed in *vacuo*. The residue was suspended in *n*-pentane (20 mL) and the liquid layer was decantated off. This procedure was repeated once. The remaining solid was recrystallized from H_2_O, yielding the product as a colorless solid, mp 117 °C (Lit. 115–117 °C [26]), yield 762 mg (4.18 mmol, 62%). TLC: *R*_f_ = 0.49 (CH_2_Cl_2_/CH_3_OH = 92:8). ^1^H NMR (600 MHz, DMSO-d_6_): *δ* (ppm) = 0.97 (d, *J* = 6.2 Hz, 3H, CHOHCH_3_), 3.61 (sext, *J* = 6.0 Hz, 1H, CHOHCH_3_), 3.72 (s, 3H, OCH_3_), 4.28 (t, *J* = 4.8 Hz, 1H, CHOHCHOHCH_3_), 4.35 (d, *J* = 5.4 Hz, 1H, CHOHCH_3_), 4.99 (d, *J* = 4.3 Hz, 1H, CHOHCHOHCH_3_), 6.82–6.88 (m, 2H, 2-H/6-H_aryl_), 7.19–7.25 (m, 2H, 3-H/5-H_aryl_). ^13^C NMR (151 MHz, DMSO-d_6_): *δ* (ppm) = 18.5 (1C, CHOH*C*H_3_), 54.9 (1C, OCH_3_), 70.6 (1C, *C*HOHCH_3_), 76.5 (1C, *C*HOHCHOHCH_3_), 112.9 (2C, C-2/C-6_aryl_), 127.9 (2C, C-3/C-5_aryl_), 135.6 (1C, C-4_aryl_), 158.0 (1C, C-1_aryl_). IR (neat): *ν* (cm^−1^) = 3305, 3248 (O–H), 2970, 2889 (C–H_alkyl_), 1609, 1512 (C=C), 1238, 1018 (C–O), 810 (C–H_aryl_). HRMS: *m*/*z* = 165.0935, calcd. 165.0910 for C_10_H_13_O_2_^+^ ([M+H-H_2_O]^+^).

## Data Availability

The data presented in this study are available in Appendix A.

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
