# Peer review of "Photocatalytic Isomerization of (E)-Anethole to (Z)-Anethole"

_molecules, 2022, doi:10.3390/molecules27165342_

Round 1
Reviewer 1 Report
The paper "Photocatalytic isomerization of (E)-anethole to (Z)-anethole" is an elegant piece devoted to the choice of the best catalyst for the E,Z-isomerization of anethole, which is known as an important precursor of several drugs. The choice of the catalyst is justified theoretically using the methods of quantum chemistry. In the experiments, the following problems are successfully solved: 1) photoreactor is constructed; 2) a reliable technique of isomers distinguishing is provided; 3) the separation of product and catalyst from the reaction mixture is achieved. It is a well-designed and thorough study, which is of use for specialists in the field of synthetic organic chemistry, especially, for those who works on the isomerization processes control.
The only minor question remains. As I know from my experience with hydrazones, E-isomers are also more stable thermodynamically, and Z-isomer can be obtained via energy expences (UV irradiation or heating; however, no catalyst required). The reverse process is spontaneous, and E-isomer is formed in couple of day. The question is: does Z-anethol returns to E-state spontaneously and, if yes, how long time it takes? Or the reversed reaction is too hindered, and can be neglected?
Author Response
Thank you very much for the revision of our manuscript.
Regarding the stability / reverse isomerization of (Z)-anethole:
Out of caution for this exact reason, the compound was stored in the freezer at -20 °C in an nitrogen atmosphere. An NMR was recorded after 3 months of storage and the determined ratio was still exactly 90 : 10.
An according sentence was added in the manuscript:
"However, the ratio of 90 : 10 did not change upon storage at ‑20 °C in nitrogen atmosphere for several months, as no isomerization of (Z)-anethole to the thermodynamically favored (E)-anethole was observed."
However we did not investigate the potential isomerization at room temperature or under the influence of light.
Reviewer 2 Report
Comments regarding the manuscript entitled: Photocatalytic isomerization of (E)-anethole to (Z)-anethole.
Authors: Marvin Korffa, Tiffany O. Paulischa, Frank Gloriusa, Nikos L. Doltsinis and Bernhard Wünsch
The subject addressed in this article is interesting and could be useful for the researchers involved in this field of science. The use of 1H-NMR techniques for the characterization of E/Z isomers could provide useful information.
Taking into consideration the observations below I recommend the publication of this manuscript after minor revisions.
The unit of measure for the ordinate (vertical) axis coordinates is missing, please specify it, (Fig2B).
Row 219 the authors mention: “methanol as a green solvent 219 (Capello et al. 2007) was used in all further experiments”. Methanol it is not considered a green solvent, reformulation is necessary.
Author Response
Thank you very much for the revision of our manuscript.
In Figure 2B the y-axis is now indicated with "Intensity".
The sentence
"methanol as a green solvent (Capello et al. 2007) was used in all further experiments"
was rephrased as:
"For reasons of sustainability, methanol was used in all further experiments, as it is the environmentally preferred solvent among the four tested (Capello et al. 2007)."
In all further cases regarding methanol, the term "green" was removed.